# Process Intensification in Bio-Ethanol Production–Recent Developments in Membrane Separation

**Izumi Kumakiri [1,*], Morihisa Yokota [1], Ryotaro Tanaka [1], Yu Shimada [1], Worapon Kiatkittipong [2], Jun Wei Lim [3], Masayuki Murata [4,5] and Mamoru Yamada [4,5]**

[1] Graduate School of Sciences and Technology for Innovation, Faculty of Engineering, Yamaguchi University, 2-16-1 Tokiwadai Ube, Yamaguchi 755-8611, Japan; morichan@yamaguchi-u.ac.jp (M.Y.); b037vf@yamaguchi-u.ac.jp (R.T.); b061vfv@yamaguchi-u.ac.jp (Y.S.)

[2] Department of Chemical Engineering, Faculty of Engineering and Industrial Technology, Silpakorn University, Nakhon Pathom 73000, Thailand; kiatkittipong_w@su.ac.th

[3] Department of Fundamental and Applied Sciences, HICoE-Centre for Biofuel and Biochemical Research, Institute of Self-Sustainable Building, Universiti Teknologi PETRONAS, Seri Iskandar 32610, Perak, Malaysia; junwei.lim@utp.edu.my

[4] Research Center for Thermotolerant Microbial Resources, Yamaguchi University, Yamaguchi 753-8315, Japan; muratam@yamaguchi-u.ac.jp (M.M.); m-yamada@yamaguchi-u.ac.jp (M.Y.)

[5] Graduate School of Sciences and Technology for Innovation, Faculty of Agriculture, Yamaguchi University, Yamaguchi 753-8515, Japan

\* Correspondence: izumi.k@yamaguchi-u.ac.jp

**Abstract:** Ethanol is considered as a renewable transport fuels and demand is expected to grow. In this work, trends related to bio-ethanol production are described using Thailand as an example. Developments on high-temperature fermentation and membrane technologies are also explained. This study focuses on the application of membranes in ethanol recovery after fermentation. A preliminary simulation was performed to compare different process configurations to concentrate 10 wt% ethanol to 99.5 wt% using membranes. In addition to the significant energy reduction achieved by replacing azeotropic distillation with membrane dehydration, employing ethanol-selective membranes can further reduce energy demand. Silicalite membrane is a type of membrane showing one of the highest ethanol-selective permeation performances reported today. A silicalite membrane was applied to separate a bio-ethanol solution produced via high-temperature fermentation followed by a single distillation. The influence of contaminants in the bio-ethanol on the membrane properties and required further developments are also discussed.

**Keywords:** bio-ethanol; thermotolerant yeast; membrane separation; ethanol-selective membrane; energy demand

## 1. Introduction

The transport sector is one of the largest contributors of greenhouse gas. Fossil-fuel is still the major energy source today and a shift to more sustainable fuels is indispensable. Huge efforts have been made to convert renewable biomass, in particular, lignocellulosic feedstocks, to biofuel. Bio-ethanol has been drawing attention among various types of biofuel, because ethanol can be mixed with petrol or used as is as a transport fuel.

Bio-ethanol production via fermentation can be divided into several stages [1], as shown in Figure 1. Pretreatment is required when using cellulosic biomass, and enzymatic saccharification is required for starchy or cellulosic biomass. Various technologies have been proposed and developed for each stage to make the conversion process more efficient and cost-effective. The usage of residue is another key to making the bio-ethanol conversion process more economically and environmentally friendly [2–4].

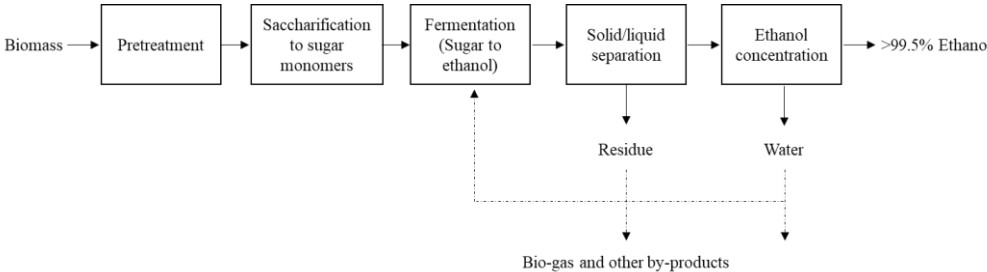

**Figure 1.** Overview of a bio-ethanol production process.

Energy consumption and distribution per each stage in Figure 1 can vary depending on the type of raw biomass, the process configuration, and other various conditions. The ethanol concentration after fermentation is low and is often less than 10%. In the case of producing a fuel grade ethanol, which requires purity over 99.5 mol% (99.8 wt%) [5,6], the fermented broth needs to be concentrated. Distillation is a well-established and widely used technology; however, when applied to anhydrous bio-ethanol production it can occupy more than one-third of the total energy consumption [7,8].

The energy demand at the conversion process should be limited as much as possible to enhance the benefit of using bio-ethanol. Developing environmentally friendly and easier operation/maintenance technologies will make the conversion process more attractive. Because bio-refineries require various technologies, collaboration efforts among different research areas are essential.

In the following, the trend in bio-ethanol demand is summarized using Thailand as an example. Then, advances in fermentation technology, comparisons of membrane integrated downstream processes with conventional distillation, and the current status of membrane properties in separating bio-ethanol are discussed. Among various technologies, this article focused on thermotolerant yeast and membrane-based technologies. The number of publications on high-temperature fermentation and bio-ethanol production using dehydration membranes has increased over the last decade, as shown in Figure 2. The literature survey was performed by Google Scholar [9] using the keywords indicated in the figure caption. Reports on ethanol-selective membranes, on the contrary, is rather small compared to the number of publications on dehydration membranes. However, a process simulation performed here showed their potential contribution to realizing increased energy-efficient bio-ethanol production. The status of ethanol-selective membranes is explained by comparing different membrane materials in the following.

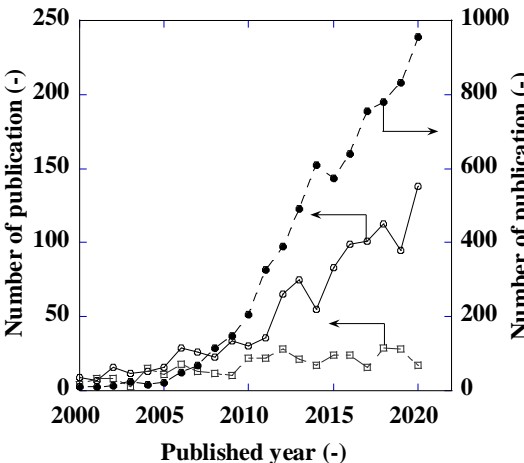

**Figure 2.** Number of publications surveyed by Google Scholar on the 3 June, 2021. The keywords used for the survey are as follows: ●, thermotolerant and bioethanol; ○, dehydration, membrane, separation, bioethanol, and hybrid process; □, ethanol selective, membrane, and separation.

## 2. Bioethanol Production in Thailand

Thailand is the seventh-largest ethanol producer, following the USA, Brazil, European Union (EU) countries, China, India, and Canada, as shown in Table 1. However, interestingly, Thailand's ethanol blend rate is the second highest ranked in the world, with an average of 13.7% in 2019, following only by Brazil, which currently mandated as high as 27% (E27), additionally with a high portion of hydrous ethanol [10].

**Table 1.** The World's top eight ethanol producers in 2019 and their blend rate in gasoline (Data taken from [11–15]).

| Country | Production (Million Liters per Day) | Blend Rate in Gasoline (vol%) |
|---|---|---|
| United State | 163.86 | 10.55 |
| Brazil | 89.40 | 27 |
| EU | 14.93 | 6.16 |
| China | 9.33 | 2.4 |
| India | 5.50 | 4.5 |
| Canada | 5.19 | 6.6 |
| Thailand | 4.36 | 13.7 |
| Argentina | 3.01 | 11.7 |

In Thailand, ethanol is initially employed to blend with gasoline as an octane enhancer for replacement of methyl *tert*-butyl ether (MTBE). It should be noted that ethanol and other alcohols produced from, e.g., in situ conversion of refinery cuts [16], were also used successfully as substitutes of gasoline ether oxygenates.

Presently, the common ethanol to gasoline blending proportion is 10%, 20%, and 85% and are referred to as E10, E20, and E85, respectively; around 66% of petrol vehicles in Thailand are compatible with E20 fuel [17]. The primary raw materials for ethanol production are molasses, cassava, and sugarcane juice [18].

Thailand's ethanol production and domestic utilization have continuously increased, as shown in Figure 3 [18,19]. The increase in gasohol consumption has resulted from government policy and subsidization from the state oil fund. However, the Thai government has adjusted the ethanol consumption target. Previously, according to the Alternative Energy Development Plan (AEDP), 2015, the target is 4.1 billion liters by 2036. In the current

plan (AEDP, 2018), the ethanol consumption target was reduced to 2.7 billion liters because the ethanol production raw materials may not be sufficient [20,21].

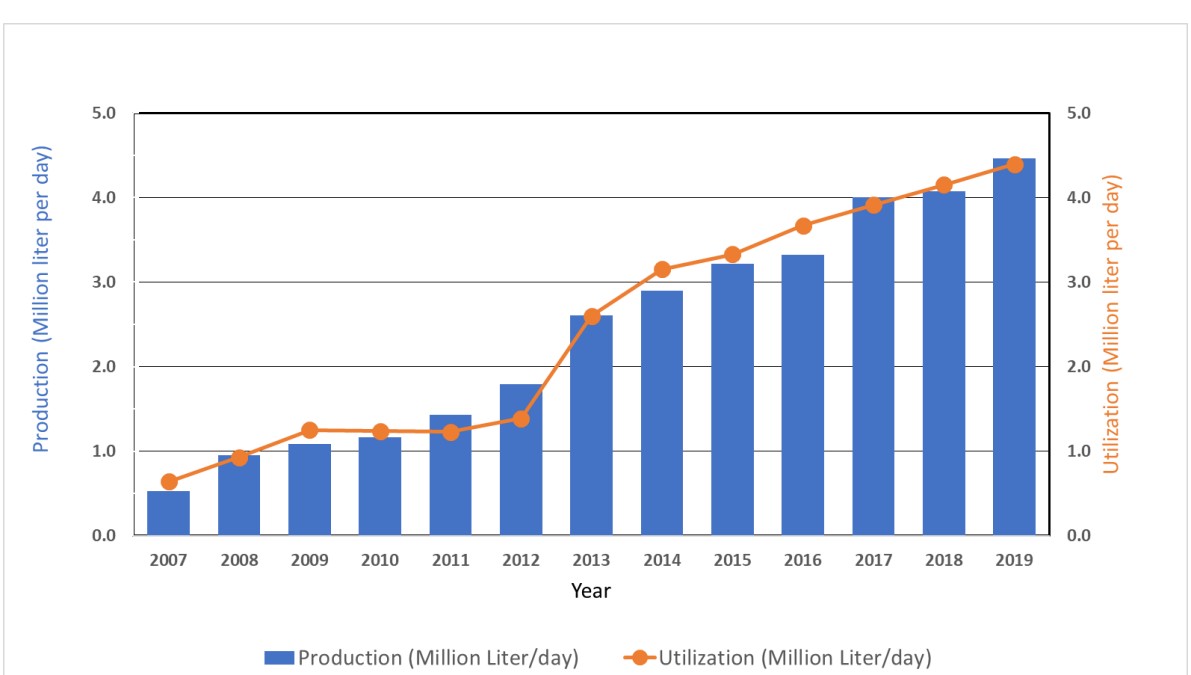

**Figure 3.** Ethanol production and consumption in Thailand (2007–2019).

Promoting ethanol production from other raw materials is necessary to overcome feedstock insufficiency problem. The recent trend for ethanol production has focused on non-edible feedstock, especially lignocellulosic materials. The Thailand Institute of Scientific and Technological Research (TISTR) has found that bagasse, rice straw, and corncobs are efficient and feasible raw materials for ethanol production at the industrial level [22]. However, the pretreatment process is the major challenge of second-generation ethanol production technology. Besides second-generation ethanol production technology, Thailand is also in search of R&D on microalgae as a third-generation ethanol production technology [23]. The support of second and third-generation ethanol production from the Thai government corresponds to the Thailand Integrated Energy Blueprint (TIEB), which sets ethanol production target from these two generations biofuels equal to 10 kilotons of oil equivalent (ktoe) by 2036 [23].

## 3. New Approaches on the Microbial Biomass Conversion

*Saccharomyces cerevisiae* is used worldwide for the fermentation of ethanol. In the conventional fermentation-distillation-dehydration process with first-generation biomass, it is not profitable unless the production scale of ethanol is 15,000 kL/year or more in Japan, based on two different demonstration projects performed in Hokkaido from 2007 to 2012. In areas where it is difficult to secure a huge amount of feedstock and the cost of transportation is high, as it is in Japan, the local production for local consumption by small-scale ethanol production is desirable. However, in order to be profitable in a small ethanol production facility, it is essential to develop an innovative technology for each step of the ethanol production process. Here, we introduce a high-temperature fermentation (HTF) technology that has several benefits, such as reduced cooling costs, reduced microbial contamination, and reduced amounts of hydrolytic enzymes used in simultaneous saccharification and fermentation (SSF) [24,25]. Fermentation is an exothermic reaction that requires cooling of the reactor, as the temperature inside the reactor rises to around 313 K. Otherwise, such high temperatures cause prevention of the fermentation ability of ethanol-producing microorganisms or cell death. A stable HTF at about 313 K that does not

require temperature control is thus highly desirable. In addition, SSF that can be performed by a relatively simple facility with easy operations is suitable for small-scale production.

On the other hand, thermotolerant microorganisms capable of efficiently fermenting and producing ethanol at high temperatures are indispensable for HTF technology. We have isolated and characterized many thermotolerant yeasts from joint research from southeast Asian countries, centering on Thailand [25]. Among them, *Kluyveromyces marxianus* DMKU3-1042, isolated in Thailand, can grow at 321 K and showed high ethanol productivity up to about 316 K when glucose was used as a carbon source. Additionally, unlike *S. cerevisiae*, *K. marxianus* utilizes a wide range of carbon sources [26] and can convert polysaccharide inulin to ethanol [27]. In addition, a *K. marxianus* strain that has a high ethanol production capacity from xylose contained in second generation biomass [28], a *Pichia kudriavzevii* strain [29] that is suitable for HTF using cassava starch as a raw material, and a *Spathaspora passalidarum* strain [30] that has no glucose suppression with high ethanol production capacity from xylose have been isolated. Other research groups have isolated thermotolerant, ethanol-fermenting yeasts from India, Bangladesh, Vietnam, and Brazil [31–34].

As mentioned above, the temperature in the reactor increases by metabolic and mechanical heat sources, being close to a non-permissible level for non-thermotolerant yeast, with *S. cerevisiae* being the most frequently applied yeast. However, thermotolerant yeasts can grow and ferment under such high temperatures, which allows us to perform temperature-non-controlled fermentation. For example, the ethanol production rate drastically decreases with non-thermotolerant yeasts when the temperature is increased from 303 to 313 K. On the contrary, the ethanol production rate is almost independent of this temperature range [26]. When a bench-scale fermentation without temperature controls using 2 L of 9% glucose medium was tested, thermotolerant *K. marxianus* DMKU 3-1042 [35,36] produced ethanol equivalent to that under the temperature-controlled condition at 303 K. Moreover, a fermenter-scale fermentation with 4000 L of 18% sugarcane was tested to achieve 7% ethanol production [37]. This fermentation is favorable because the cooling cost tends to be higher in tropical countries or increases in summer in many other countries. In addition, temperature-non-controlled fermentation can keep producing ethanol, even if any power outage occurs. As HTF reduces the risk of undesirable microorganisms growth, the air-lock of the fermenter does not need to be perfect, which makes the fermenter design simpler.

Another interesting advantage of HTF is the possibility to combine fermentation and vacuum distillation in one unit. Such a process can reduce the manufacturing time and the cost of equipment. The higher fermentation temperature increases the saturated vapor pressure of ethanol. The higher saturated vapor pressure facilitates the vapor distillation that requires less pressure than the saturated vapor pressure. A system consisting of a fermentation and a distillation tank, the primary and secondary ethanol recovery units, a vacuum pump, and a drain unit was constructed [37]. Ethanol is concentrated as the process proceeds from the primary to the secondary ethanol recovery unit, and the air in the tank is discharged outside during the vacuum distillation; some ethanol is trapped in the drain unit. When fermentation with *K. marxianus* DMKU 3-1042 and distillation at 70 mbar and 314 K were applied, about 35% and 60% were recovered in the primary and secondary bottles, respectively [37]. The process of the simultaneous fermentation and distillation under low pressure was continuously repeated three times with 12% rice-hydrolysate. There are some additional benefits in this system: (a) microbes avoid exposure to high concentrations of ethanol or acetic acid, or strong oxidative stress and (b) fermentation can be continued during distillation, increasing ethanol yields. The fermentation and vacuum distillation combined system removes solid materials in the fermented broth from the liquid fraction. The ethanol can be concentrated further by applying membranes or other separation technologies.

### 4. Various Membrane Separation Processes

Employing distillation is a conventional way to increase the ethanol concentration after fermentation. Azeotropic distillation is required to obtain over 96 wt% (89 mol%) ethanol. Adsorption columns can be used to produce anhydrous ethanol [38]. Recently, a hybrid process consisting of distillation and membrane dehydration was proposed as an energy-saving alternative. In the hybrid process, azeotropic distillation was replaced with membrane separation [39–41]. In the late 1990s, a successful industrial application of A-type zeolite membranes, a type of inorganic membrane, to dry solvents was reported [42]. Since then, the number of industrial applications of membrane integrated processes has been growing, and more than two hundred units are under operation today.

Energy consumption in bio-ethanol production may be reduced further by employing membranes in different ways. Various membrane processes, including pervaporation (PV) [42], membrane distillation (MD) [43,44], vapor permeation (VP) [41,45], nano-filtration (NF) [46], reverse osmosis (RO) [47], and forward osmosis (FO) [48,49] have been intensively studied to concentrate liquid mixtures, together with the developments of new types of membranes. Membranes can be integrated with saccharification or fermentation to concentrate sugars and remove fermentation inhibitors [48,49]. Membranes can also be used to concentrate ethanol after fermentation. In the latter case, dehydration PV membranes have been studied the most. While a phase change occurs during the permeation through PV and MD membranes, NF and RO processes are a pressure-driven separation with no phase change. Considering the large latent heat of water, NF and RO can be more energy-efficient than PV.

Permeation equations, for example a solution-diffusion model, relate these different membrane processes [43,50]. Nakao compared the PV and RO processes using the transport model and proposed a combination of water-selective RO membranes and ethanol-selective RO membrane [51], as shown in Figure 4. The energy requirement to concentrate 10% ethanol to 96% with ethanol recovery over 95% was studied. The study showed that the RO process is much more energy efficient than PV and requires 1/1000 of the energy required by distillation.

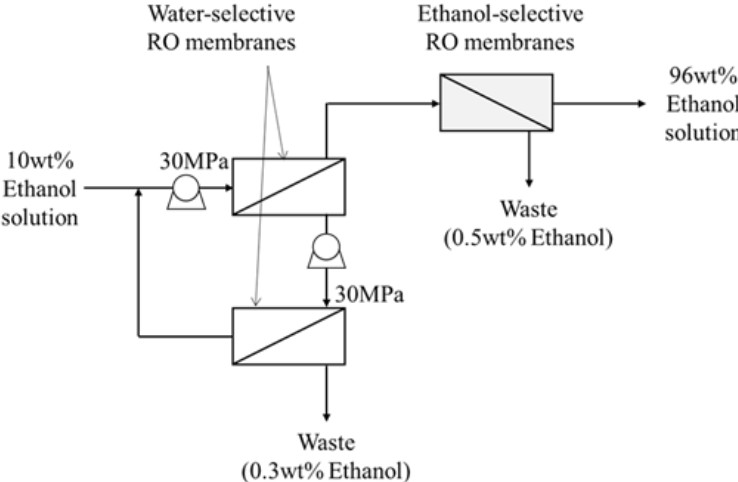

**Figure 4.** Example of combining two types of RO membranes (modified from [51]).

The RO-based separation process is with a simple design as illustrated in Figure 4. The major components are pumps to raise the feed side pressure, membrane modules and valves to regulate pressure and flow rate., A heating or cooling system is dispensable. One of the drawbacks of RO is the high pressure required at the feed side to overcome the osmotic pressure across the membrane. Conventional polymeric membranes are difficult to apply under the high pressure of 30 MPa assumed in the above process calculations

[51]. Therefore, inorganic membranes have been investigated. One of the first attempts at an inorganic RO membrane is the application of an A-type zeolite membrane to water/ethanol separation [47]. The membrane showed pressure stability at least up to 5 MPa and water-selective permeation, but the flux was too small. Recently, various types of inorganic RO membranes have been developed for desalination purpose [52,53]. However, further developments are required to realize a RO-membrane based downstream process for bio-ethanol production.

Besides dehydration VP/PV membranes [42,54], there are membranes permeating more ethanol than water. Figure 5 shows the liquid-gas equilibrium of an ethanol/water mixture. Examples of permeate concentration through membranes as a function of ethanol concentration in the feed solution are added to the figure. A-type zeolite [54] and Silicalite [55] membranes were used as examples of water- and ethanol-selective membranes, respectively. Dashed lines in the figure show an example of the first tray composition in distillation. Several trays are required to concentrate, e.g., 10% ethanol to over 80%, by distillation. On the contrary, over 80% ethanol can be obtained after a single permeation through a hydrophobic membrane [55–59]. Membrane separation is not limited by the azeotrope. Water-selective membranes permeates over 99.9% water for a large ethanol concentration range in the feed. In the following, a few configurations using water-selective and ethanol-selective VP/PV membranes are compared with conventional distillation.

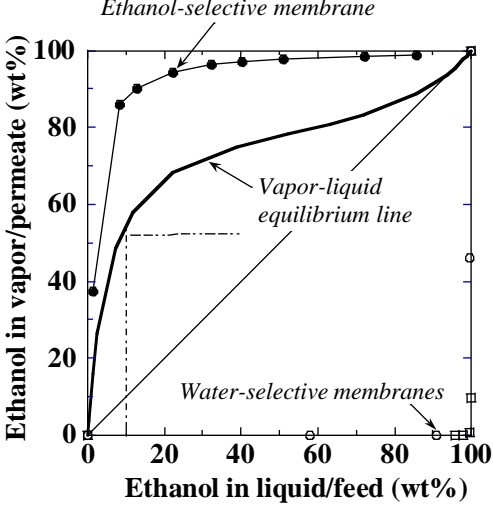

**Figure 5.** Liquid-vapor equilibrium of ethanol-water mixture with example separation performances of water- and ethanol-selective membranes: solid line, liquid-gas equilibrium; dashed lines, examples of stages in the distillation column; ●, ethanol concentration in the permeate of an ethanol-selective silicalite membrane [55], □; ○, ethanol concentration in the permeate of water-selective A-type zeolite membranes (□, [54]; ○, this study).

## 5. Combination of Distillation and Water-Selective VP/PV Membranes

A hybrid process combining distillation and membrane dehydration is getting increasingly accepted as an energy-saving alternative to concentrate close-boiling mixtures, such as ethanol solution and iso-propyl alcohol solution [41,60]. In this hybrid process, the feed to the membranes can be a liquid mixture (PV) [61] or a vapor mixture (VP) [41]. In some of the industrially applied hybrid process using A-type zeolite membranes, a fraction of vapor from the top of a distillation tower is fed to the membrane. If no heat is added to the membrane unit, the fluid may be liquefied while flowing over the membrane. While A-type zeolite membranes show quite high water-selective separation properties in both PV and VP separations, the application of this membrane is limited to dry solvents with

less than 15% water due to the insufficient stability of the membrane in water-rich conditions. Recently, various other types of zeolite membranes, such as zeolite T, CHA, MOR and ZSM-5, having higher stability in water-rich and in acidic media have been reported [62–64]. The application of dehydration membranes in the hybrid process can be extended to fluids containing higher amounts of water with these new types of membranes. However, the impact of extension on the total energy consumption, process size, and other factors are not clear.

The energy consumption and ethanol recovery rate were compared by changing the inlet ethanol concentration to a dehydration membrane. Concentrating 10% ethanol to 99.5% was assumed in the calculation using a cape-open to cape-open (COCO) program [59]. Figure 6 shows some of the process schemes compared in this study. Distillation followed by azeotropic distillation was used as a standard case (Figure 6a). The downstream process can be combined with continuous fermentation, as illustrated in the figure. Figure 6b shows a schematic view of the hybrid process, where azeotropic distillation is replaced with membrane dehydration. The membrane operational temperature was assumed to be the same as the distillation tower top temperature. A vapor mixture was fed to the membrane in this assumption. Neither membrane flux nor required membrane area were considered. The membrane separation factor was assumed to be 2000, which means that 0.05% ethanol in the feed mixture permeates through the membrane. A sweep gas was applied to the permeate side of the membrane instead of the vacuum lines often used in the VP/PV tests [50,55,62]. The sweep gas flow was assumed to be nine times higher than the total flux through the membrane. The outlet of the sweep gas was emitted to the atmosphere in the calculation, which reduced the ethanol recovery.

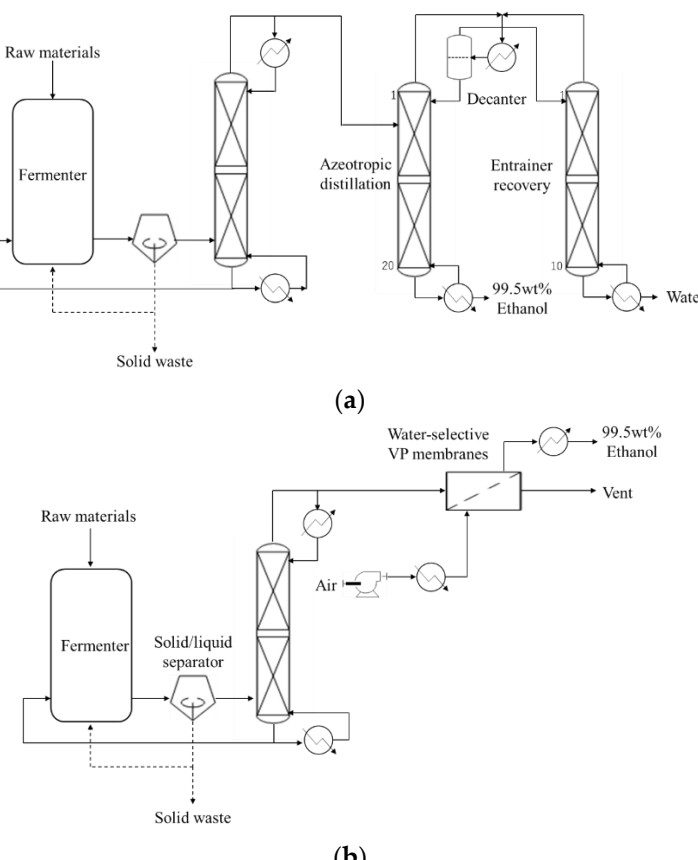

**Figure 6.** Distillation and dehydration membrane combined process: (**a**), diagram of distillation followed by azeotropic distillation; (**b**), diagram of distillation and membrane dehydration with sweep system combined.

With the above assumptions, azeotropic distillation required 2339 W/kg-ethanol to concentrate 80% ethanol to 99.5%. On the contrary, the energy consumption by membrane dehydration was 287 W/kg-ethanol, which is one order smaller than that of distillation. A significant reduction in the separation energy by replacing azeotropic distillation with membrane dehydration was also reported with different process configurations and assumptions [60]. The calculation showed that the dehydration membrane has a significant advantage, even with applying sweep gas at the permeate side.

Lowering the ethanol concentration at the inlet of a dehydration membrane reduces the distillation load. For example, changing the inlet ethanol concentration from 85 to 75 wt%, the energy demand at the distillation became 1820 to 1790 W/kg-EtOH. On the contrary, lowering the ethanol concentration in the feed to a membrane enhances the membrane duty. As the outlet ethanol concentration is fixed to 99.5%, the total amount of water permeating through a membrane increases when the feed ethanol concentration is smaller. Accordingly, a higher sweep flow rate is required to maintain the pressure difference of water across the membrane, which increased the compressor energy and the energy to heat the sweep gas to the membrane operation temperature. As a result, the energy demand by changing the inlet concentration to the membrane became minor. For example, reducing the ethanol concentration at the inlet of the membrane unit from 85 to 75% changed the total energy demand of the hybrid process to concentrate 10% ethanol to 99.5% from 1840 to 1830 Wh/kg-EtOH. The impact of extending the application of the dehydration membrane is not significant as compared with the replacement of azeotropic distillation with membrane dehydration under the process configuration and the assumptions used in this study.

## 6. Combination of Ethanol-Selective and Water-Selective VP/PV Membranes

Hydrophobic membranes selectively permeate ethanol over water. Table 2 shows some examples of ethanol-selective membranes, which can be divided into polymeric membranes, polymer and filler composite membranes, which are called mixed matrix membranes (MMMs), and inorganic membranes. Separation factor, $\alpha$, is defined as:

$$\alpha = (y_{Etahnol}/y_{Water})/(x_{Ethanol}/x_{Water}) \tag{1}$$

where x and y are the mass fraction of each component in the feed and in the permeate, respectively. Permeate concentrations in the table were calculated from the separation factor and feed composition given in the references. Polymeric membranes, including commercially available membranes, showed slightly higher ethanol-selectivity than the liquid-vapor equilibrium [65–67]. The ethanol-selectivity can be enhanced by mixing hydrophobic fillers to the polymers (MMMs) [65,67,68]. Inorganic membranes, consisting of pure filler materials, in the table showed a higher ethanol-separation ability with higher flux [55–57]. Both silicalite and beta-zeolite membranes are types of zeolite membranes. These membranes are made of pure silica and have ordered pores of sub-nanometers. The ethanol-selective permeation is based on the strong adsorption of ethanol to the zeolitic pores, which inhibits the water permeation. Beta-zeolite has larger pores than silicalite. The pores may be too large to be plugged by adsorbed ethanol and let some water permeating through, which results in a lower ethanol-selectivity in beta zeolite membranes than in silicalite membranes. The reported flux values of silicalite membranes have some variations as the membrane micro-morphologies depend on the preparation conditions, such as the support types used and hydrothermal synthesis conditions. Nevertheless, several groups reported that when silicalite membranes were applied to ca. 10% ethanol solution, the ethanol concentration in the permeate was over 80% [55,56,58]. Based on these results, the separation factor of ethanol-selective membrane was assumed to be 36 in the following simulations.

**Table 2.** Examples of ethanol-selective membranes.

| Membrane Type | Membrane Material | Feed Ethanol Conc. (wt%) | Temperature (K) | Permeate Ethanol Conc. (wt%) * | Separation Factor, $\alpha$ (-) | Flux (Kg m$^{-2}$ h$^{-1}$) | Ref. |
|---|---|---|---|---|---|---|---|
| Polymeric membranes | PDMS [+] (Pervatech, Netherland) | 5 | 323 | 26 | 6.7 ± 1.0 | 2.6 ± 0.4 | [66] |
| | PERVAP 4060 (Sulzer Chemtech, Switzerland) | 5 | 323 | 27 | 7.1 ± 1.3 | 1.3 ± 0.3 | [66] |
| | PERVAP 4060 (Sulzer Chemtech, Switzerland) | 5 | 303 | 32 | 9 | 0.6 | [67] |
| Mixed matrix membranes (MMMs) | PDMS-silicalite hollow spheres (30 wt% **) | 6 | 313 | 49 | 15.3 | 0.07 | [68] |
| | PDMS-hydrophobized Al$_2$O$_3$ [+] (1 wt% **) | 5 | 303 | 37 | 11 | 0.06 | [67] |
| | PTMSP [#] -silica (1.5 wt% **) | 10 | 323 | 63 | 15.3 | 0.40 | [65] |
| Inorganic membranes | Silicalite | 12 | 333 | 89 | 58 | 0.76 | [55] |
| | Silicalite | 10 | 323 | 91 | 92 | 3.00 | [56] |
| | All silica beta-zeolite | 10 | 323 | 58 | 12.3 | 6.29 | [57] |

* calculated from the feed concentration and the separation factor given in the paper, ** amount of filler mixed to the polymeric matrix, [+] surface modification using triethoxy(octy)silane, [†] PDMS: polydimethylsiloxane, Silicalite: all-silica MFI type zeolite. [#] PTMSP: poly(1-trimethylsilyl-1-propyne).

Combining ethanol-selective and water-selective VP/PV membranes, it is possible to eliminate the distillation column completely. Figure 6 shows schemes of a downstream process employing these two types of membranes. An ethanol-selective membrane was applied to concentrate 10% ethanol to 80%, then a water-selective membrane was applied to dehydrate ethanol to 99.5%. Differently from water-permeating membranes described in the above section, the ethanol vapor permeating through the first membrane unit needs to be collected. As industrial water with a temperature of 300 K was not sufficient to condense ethanol vapor, a chiller of 253 K with 50% efficiency was added in the calculation. Two configurations were considered for the permeate side of the ethanol-selective membrane: a depressurized system (Figure 7a) and a sweep air system (Figure 7b). The operation temperature of both ethanol- and water-selective membranes was assumed to be 337 K. The stage cut-off, the fraction of feed permeating through a membrane, was assumed to be 90%. The retentate of feed was recycled to the fermenter, as shown in the figure. The ethanol recovery rate was calculated by considering the ethanol loss at the liquid/gas separator and through the water-selective membrane.

Table 3 shows the energy required to concentrate 10% ethanol to 80%, calculated with the process scheme in Figure 7. In the case of the depressurized permeate system, as shown in Figure 7a, the condensation of ethanol vapor was not possible with industrial water, which increased the duty on the chiller. As a result, the total energy requirement per kilogram of ethanol was almost equivalent to the distillation case. On the contrary, in the compressed air sweep system, as shown in Figure 7b, about 20 to 35% energy reduction was obtained from the distillation energy demand. Higher sweep gas flow rate requires more energy as it increases the compression energy, the heat required to bring the sweep gas to the membrane temperature, and the chiller energy to condense permeated vapor with the sweep gas. Too small sweep flow rate will limit the membrane flux as the driving force of the permeation gets smaller [50]. Accordingly, there is a range of optimum flow rate, which needs further investigation. In the case with a sweep gas, a liquid/gas separator is required after the chiller, as shown in Figure 7. The separator of liquid from gas reduced the ethanol recovery. About 0.5 to 1.3% of the ethanol permeated through the ethanol-selective membrane was lost at the separator, as some ethanol vapor was removed with the exhaust gas. The loss depends on the sweep gas flow rate; the higher the flow rate, the larger the loss. While an ethanol-selective membrane with a sweep flow system has the possibility to reduce energy consumption, the optimum process configurations need further study.

**Table 3.** Comparison of the energy required to concentrate 10 wt% ethanol to 80 wt%.

| Process Configuration | | Energy Demand (W/kg-EtOH) | | |
|---|---|---|---|---|
| | | Feed Treatment | Chiller | Total |
| | Distillation | - | - | 1803 |
| Membrane separation | Vacuum at the permeate side | 906 | 956 | 1862 |
| | Air sweep at the permeate side (x 1.3 *) | 930 | 270 | 1199 |
| | Air sweep at the permeate side (x 2.2 *) | 946 | 379 | 1325 |
| | Air sweep at the permeate side (x 3.1 *) | 962 | 488 | 1450 |

* Sweep flow rate compared to the flux through a membrane.

Various process configurations are possible to concentrate 10% ethanol solution to 99.5%. A few conditions are compared in Table 4. Replacing azeotropic distillation with membrane dehydration reduces the energy required for the separation to about half. Employing ethanol-selective membranes with compressed air as a sweep gas at the permeate side instead of distillation can further reduce the downstream energy. A membrane-based downstream process is an interesting choice, especially for small to medium scale applications, where the scale-merit of distillation is limited. The modular design of membrane units and the simple operation/maintenance are other advantages of the membrane process. In addition, membrane separation is environmentally friendly as no or very few additional chemicals are required [69].

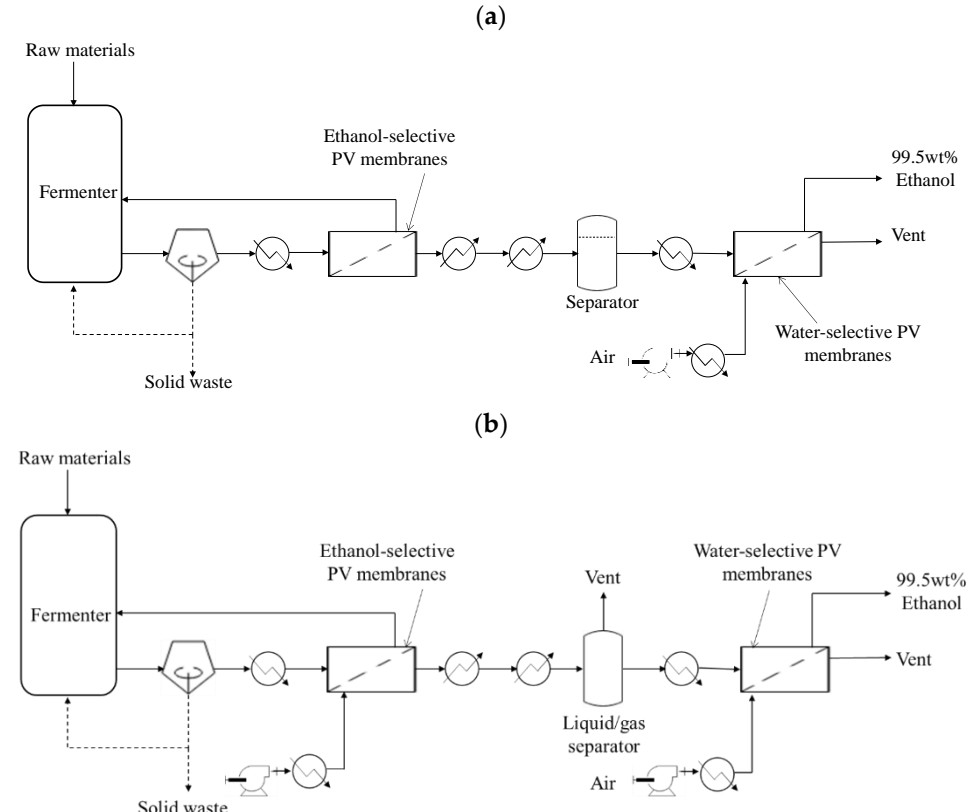

**Figure 7.** Downstream process consisting of two types of membranes: (**a**), a diagram of ethanol-selective membrane with vacuum followed by water-selective membrane with sweep; (**b**), diagram of ethanol- and water-selective membranes with sweep.

**Table 4.** Comparison of the energy demand to concentrate ethanol from 10 to 99.5 wt%.

| Process Configuration | Energy Demand (W/kg-EtOH) | | | Ethanol Recovery (%) |
|---|---|---|---|---|
| | 10 wt% Ethanol to 80 wt% | 80 wt% Ethanol to 99.5 wt% | Total | |
| Distillation +azeotropic distillation | 1804 | 2339 | 4142 | 99.95 |
| Distillation + water-selective membrane [#] | 1804 | 287 | 2091 | 99.95 |
| Ethanol-selective membrane with vacuum at the permeate side + water-selective membrane [#] | 1862 | 287 | 2149 | 99.5 |
| Ethanol-selective membrane with 1ir sweep at the permeate side (×1.3 [#]) + water-selective membrane * | 1199 | 277 | 1480 | 99.4 |
| Ethanol-selective membrane with 1ir sweep at the permeate side (×2.2 [#]) + water-selective membrane * | 1325 | 277 | 1602 | 99.0 |
| Ethanol-selective membrane with 1ir sweep at the permeate side (×3.1 [#]) + water-selective membrane * | 1450 | 278 | 1728 | 98.7 |

* Sweep at the permeate side with flow rate 9.1 times higher than the membrane flux, [#] Sweep flow rate compared to the flux through a membrane.

One of the current challenges is the higher cost of inorganic membranes compared to conventional polymeric membranes. It is difficult to find the exact price of membranes as it depends on the production method, scale, and other factors. However, a few have reported that the inorganic membrane cost is about two orders higher than the polymeric membrane [70,71]. The robustness of the inorganic membranes may give the membranes competitiveness in long-term usage; however, stabilities of the membranes need to be evaluated under realistic conditions. The ceramic support cost occupies a major part of the fabrication cost of zeolite and zeolite-like micro-porous inorganic membranes [70]. Therefore, various ideas, using, for example, less expensive hollow fiber supports [71], reusable stainless-steel supports [72], and other materials [73,74] are proposed. Improving the membrane flux is another approach because higher flux requires less membrane area and will reduce the capital cost of the membrane unit. The membrane synthesis conditions should be scalable in an economic and environmentally friendly way. The successful industrialization of dehydration zeolite membranes and modules [42,54] gives a guideline for the implementation of ethanol-selective membrane processes.

## 7. Ethanol-Selective Silicalite Membranes Applied to a Fermented Solution

Membrane properties are often evaluated with a single component permeation or a separation of binary synthetic mixtures. For example, the performances in Table 3 were measured with synthetic ethanol/water mixtures. On the contrary, bio-ethanol produced via fermentation contains various components. The influences of these third chemicals are not possible to predict at this moment. Nomura et al. studied the influence of yeast and salts in ethanol solutions. They reported that salts enhanced the ethanol selectivity due to the salt effect on the liquid-vapor equilibrium that enlarged the ethanol vapor pressure [58]. Other contaminants than salts, such as acids, can also influence the membrane properties. Offeman et al. used mixed matrix membranes consisting of hydrophobic zeolite particles and polydimethylsiloxane (PDMS) and applied them to separate fermentation broths [75]. They reported reductions of selectivity and flux in the broth due to a strong adsorption of, e.g., oleic acid to the zeolitic pores.

Figure 8 shows the flux and the concentrations of feed and permeate through a silicalite membrane. A distillate after a single distillation of fermented broth obtained by a HTF was used as feed. In this case, only volatile contaminants co-exist in the aqueous ethanol solution. The distillate had an ethanol concentration of 14 wt% and a pH of about 4. A synthetic mixture of 14 wt% ethanol solution was applied before and after the distillate test to check the change of membrane performance. All the tests were performed at 348 K. The selectivities of the silicalite membrane, measured with a synthetic mixture before and after the distillate separation, were the same, suggesting that the selective layer was stable. On the contrary, the flux became about half after the distillate separation. Both

selectivity and flux decreased when a distillate was used as feed instead of a synthetic mixture. Acids or other volatile components in the fermented broth may be adsorbed into the membrane surface and block the zeolitic pores. The results suggested a contribution of reversible adsorption and irreversible changes. Pre-treating the fermented broth to reduce the concentration of contaminants, or modifying the membrane surface to prevent adsorption of contaminants, may reduce the reduction of the membrane property in a bio-ethanol separation. Another aspect of the ethanol-selective membrane process is the need to develop a cost-effective, large-scale membrane preparation method.

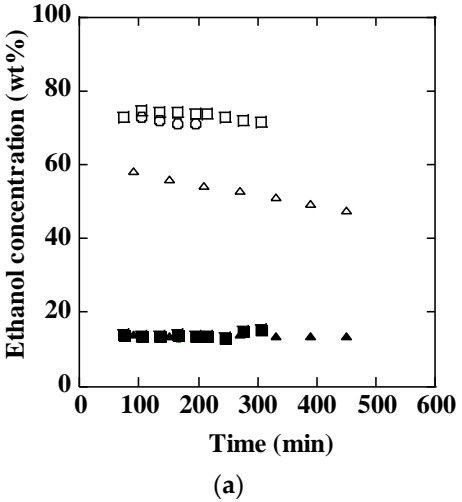

(**a**)

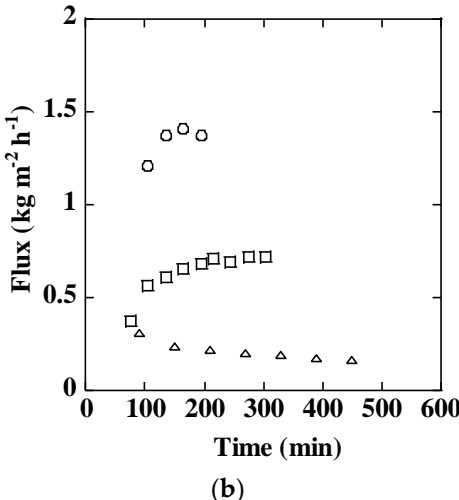

(**b**)

**Figure 8.** Changes of silicalite membrane permselectivity with time in a synthetic mixture and in a distillate of fermented broth measured at 348 K: (**a**), ethanol concentration in the feed and in the permeate; (**b**), flux as a function of time; ○, a synthetic mixture of 14 wt% ethanol as feed measured before the distillate separation; △, distillate as feed; □, a synthetic mixture of 14 wt% ethanol as feed measured after the distillate separation; open keys, permeate concentration and flux; closed keys, feed concentration.

## 8. Conclusions

To meet the increasing demand and to improve the profit of using bio-ethanol, energy consumption at the conversion process should be limited. High-temperature fermentation (HTF) does not require temperature control. Cooling water is dispensable, making the fermenter configuration simpler and the energy demand smaller. This type of fermentation is particularly interesting in tropical countries.

Membranes can be integrated into the conversion process in different ways. Dehydration PV membranes are getting increasingly accepted as an energy-saving alternative to azeotropic distillation. To simplify the membrane separation process, replacing the vacuum line at the permeate side with a compressed air flow was considered. A preliminary simulation showed that the energy demand can also be reduced by applying water-selective VP membranes with sweep gas.

The preliminary simulation also showed that employing ethanol-selective membranes with sweep gas instead of distillation can reduce about 20–30% of the energy demand to concentrate 10 wt% ethanol to 80 wt%. Several research groups have reported that silicalite membranes, a type of nano-porous inorganic membrane, can concentrate 10 wt% ethanol to over 80 wt%. However, in this study, when a silicalite membrane was applied to a bio-ethanol solution produced by HTF followed by a single distillation, the flux through the membrane became about 10% of the flux obtained with an ethanol/water synthetic mixture. This result shows that it is necessary to pre-treat the bio-ethanol before applying a membrane or develop new types of membrane whose adsorption is influenced less by contaminants.

Combining HTF and membrane separation has the potential to realize a simple conversion process, which will facilitate on-site bio-ethanol production at farm areas. However, further developments in each of the technologies, investigation of a better configuration of the integrated process, and a scale-up study are required.

**Author Contributions:** Conceptualization, I.K.; methodology, M.Y. (Morihisa Yokota), R.T. and Y.S.; writing—original draft preparation, I.K., W.K., M.M. and M.Y. (Mamoru Yamada); writing—review and editing, I.K., W.K. and J.W.L.; funding acquisition, I.K., M.M. and M.Y. (Mamoru Yamada). All authors have read and agreed to the published version of the manuscript.

**Funding:** This research was funded by e-ASIA Joint Research Program, which was granted by Japan Science (JPMJSC16E5).

**Institutional Review Board Statement:** Not applicable.

**Informed Consent Statement:** Not applicable.

**Data Availability Statement:** Not applicable.

**Conflicts of Interest:** The authors declare no conflict of interest. The funders had no role in the design of the study; in the collection, analyses, or interpretation of data; in the writing of the manuscript, or in the decision to publish the results.

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
