# Peer review of "Process Intensification in Bio-Ethanol Production–Recent Developments in Membrane Separation"

_processes, doi:10.3390/pr9061028_

Round 1
Reviewer 1 Report
The current manuscript deals with bioethanol production process and emphasizes in the use of membranes in the ethanol recovery. The paper could be characterized as a narrative review. Authors used adequate number of relating references. Moreover, the discussion upon the referred results is accurate and to the point. However, authors ought to take into serious consideration the following revisions:
- There are no available info regarding the methodology for choosing the including references (e.g. inclusion criteria, exclusion criteria, keywords used, time period of the review etc.).
- A PRISMA template would be helpful indeed for organizing the current review article.
- 69-70: It would be important to mention that ethanol and other alcohols were also used successfully as substitutes of GEOs (Gasoline Ether Oxygenates) with in situ conversion of refinery cuts (https://doi.org/10.1016/j.cattod.2014.07.058).
Author Response
We wish to thank the reviewers for the constructive comments. The comments provided valuable insights to refine the contents. We have modified the manuscript based on the comments. Replies to the comments and changes are indicated in blue in the following. We sincere hope these changes are agreeable.

Reviewer 2 Report
The topic of the article is very relevant, especially in the view of the modern decarbonization policy in the economy and alternative energy. Bioethanol is gaining attention among various biofuels because ethanol can be blended with gasoline or used as a transport fuel. The combination of high-temperature fermentation and membrane separation could be utilized in a simple conversion process that facilitates the production of bioethanol. The article defines the vector of development of each of the investigated technologies, including the study of the best configuration of the integrated process. However, before publication there are still a few comments for author to be paid attention to:
- The authors said that high-temperature fermentation is of particular interest in tropical countries. It would be nice to add some data about the features of the use of such fermentation in the tropics, for example, thermodynamic, average humidity and other parameters that affect production conditions.
- It would be appropriate to supplement Table 1 with the world leaders in ethanol production and blend rate, since the text provides a comparison with Brazil, and Table 1 only consists of Asian countries.
- The data in Table 2 duplicates the data in Table 5. The information may need to be restructured.
- In figure 6 only diagrams "a" and "b" are presented, and the figure caption notes diagram "c", which is probably a mistake.
- The first paragraph on page 3 contains the AEDP data but does not provide relevant links and the first two sentences of the Item 3 also contain information without references to any specific articles. It would be nice to add corresponding references.
6. It would be nice to style every reference in the article to a single format. For example, consider using [10-14] instead of [10] [11] [12] [13] [14] on line 67, and separating the word and reference with a space on lines 86, 106, and others.
Author Response

(The authors gave the same response as above.)

Reviewer 3 Report
In my opinion, the manuscript ID processes-1253234 entitled “Process intensification in bio-ethanol production - recent developments in fermentation and membrane separation” is interesting and valuable. Corresponds to the Prcesses journal profile. It is a very rich compendium of knowledge on current bio-ethanol separation technologies.
In my opinion, the title of the manuscript is misleading because the authors devote little space to the fermentation process. This can only be said briefly about the directions of development of the alcoholic fermentation process and current trends, mainly high-temperature fermentation. Therefore, I believe that the title of the manuscript should be reduced to "Process intensification in bio-ethanol production - recent developments in membrane separation". Another solution is to expand the section on fermentation processes.
The Authors in the title should also take into account the fact that the described scope of bio-ethanol separation technology is based on solutions in Thailand.
I believe that while the processes and technological parameters of membrane separation have been presented in an appropriate manner, more attention should be paid to economic, economic and environmental aspects. Therefore, it is necessary to refer to the LCA (life cycle assessment) and LCC (life cycle costing) analyzes existing in the literature. This is necessary in order to fully evaluate the technologies available and characterized in the manuscript.
Author Response

(The authors gave the same response as above.)

Round 2
Reviewer 2 Report
This version of the article can be recommended for publication